# Closing yield gaps for rice self-sufficiency in China

Nanyan Deng [1], Patricio Grassini [2], Haishun Yang[2], Jianliang Huang [1], Kenneth G. Cassman [2] & Shaobing Peng [1]

China produces 28% of global rice supply and is currently self-sufficient despite a massive rural-to-urban demographic transition that drives intense competition for land and water resources. At issue is whether it will remain self-sufficient, which depends on the potential to raise yields on existing rice land. Here we report a detailed spatial analysis of rice production potential in China and evaluate scenarios to 2030. We find that China is likely to remain self-sufficient in rice assuming current yield and consumption trajectories and no reduction in production area. A focus on increasing yields of double-rice systems on general, and in three single-rice provinces where yield gaps are relatively large, would provide greatest return on investments in research and development to remain self-sufficient. Discrepancies between results from our detailed bottom-up yield-gap analysis and those derived following a top-down methodology show that the two approaches would result in very different research and development priorities.

[1] National Key Laboratory of Crop Genetic Improvement, MARA Key Laboratory of Crop Ecophysiology and Farming System in the Middle Reaches of the Yangtze River, College of Plant Science and Technology, Huazhong Agricultural University, Wuhan 430070 Hubei, China. [2] Department of Agronomy and Horticulture, University of Nebraska-Lincoln, PO. Box. 830915, Lincoln, NE 68583-0915, USA. Correspondence and requests for materials should be addressed to S.P. (email: speng@mail.hzau.edu.cn)

China must make strategic decisions about how to ensure food security for 1.4 billion people, and those decisions will have a large impact on global agriculture and land use. Currently self-sufficient in rice production, China's most important food crop, the annual production of about 206 million metric tons (MMT) represents 28% of global rice supply[1,2]. At issue is whether agricultural policy should target continued self-sufficiency in rice or accept becoming a major rice importer. Resolution of this issue will markedly influence global rice markets because reliance on imports for just 10% of China's rice consumption represents more than 35% of all internationally traded rice based on 2013–2015 global export statistics[3].

Current self-sufficiency has been achieved by raising rice yields more than 50% since 1980 despite an 11% reduction in harvested rice area[1]. But the rate of rice yield growth has slowed markedly in recent years[4], and prime farmland and water for irrigation are becoming increasingly scarce as the rural-to-urban demographic shift drives fierce competition for both land and water[5,6]. Given the prognosis for little increase, or even a reduction, in rice production area, making an informed decision about whether to pursue continued self-sufficiency depends on the potential for increasing yields on existing rice area. Yield-gap analysis provides the means for estimating this untapped production potential by estimating the difference (i.e., gap) between current farm yields and the potential yield that can be achieved when yield losses from nutrient deficiencies, pests, and diseases are minimized[7].

Robust estimation of rice yield gaps in China is complicated by the large variation in production systems and climates in which rice is grown, from warm sub-tropics at 18° N latitude to cool temperate climates at 50° N. Potential yield is a location-specific property because it depends on the local weather and crop growth duration. In the irrigated systems that dominate rice production in China, location-specific factors governing potential yield include the length of growing season, determined by temperature regime, and the amount of light intercepted by the leaf canopy during the crop growth period, determined by incident solar radiation, leaf area development, and persistence[8]. Hence, accurate yield gap estimation requires good quality, long-term weather records, and data on current crop yields and management practices with adequate spatial resolution to support simulation of potential yield across the large environmental variation that characterizes Chinese rice production[9]. To date, it has not been possible to perform such a detailed analysis due to lack of both a spatially explicit dataset on rice production systems across this wide range of environments, and a suitable upscaling technique for aggregating results to a national scale.

To fill this void, we report a spatially explicit yield-gap analysis of Chinese rice production using primary data and bottom-up scaling methods recently developed for the Global Yield Gap Atlas (GYGA) (www.yieldgap.org)[9,10]. In this yield-gap analysis, we evaluate future scenario options based on estimated rice production capacity on current Chinese rice area, identifying specific regions and rice production systems that deserve highest priority for research and development investments to achieve greatest rice production on a limited supply of prime farmland. We find that a focus on increasing yields of double-rice systems in general (i.e., two rice crops per year planted and harvested in the same field), and on three provinces where yield gaps are relatively large and single-rice systems predominate (i.e., one rice crop per year on a given field), would provide greatest return on investments in research and development. Discrepancies between results from our detailed yield-gap analysis and those derived following a top-down methodology indicate that the two approaches would result in very different research and development priorities.

## Results

**Recent trends in rice cropping systems**. The geography of Chinese rice production has undergone enormous changes over the past 35 years. Whereas double-rice systems that dominate in the warm climates of central-south and south coastal regions accounted for 66% of national total harvested rice area in the 1980s, they currently represent less than 40% (Fig. 1). In contrast, area given to single-rice systems in cooler climates of central and northern regions has increased steadily although this expansion did not overcome the reduction in double-rice area. Hence, total harvested rice area has decreased by about 3.6 million hectares (Mha) since 1980. The main reasons for decrease in double-rice were (i) rapid urbanization in south coastal and central-south regions and associated conversion of rice land to housing, industry, and supporting infrastructure, and (ii) decreased rural labor availability and rising labor costs leading to lower net income than for single-rice systems, which require less labor[11]. Taken together, trends in production area and yields have resulted in the dominance of single-rice systems, which now account for more than 65% of national rice production (Fig. 1).

**Current potential yield and yield gaps**. To achieve the required level of spatial resolution for robust estimation of irrigated rice yield gaps requires primary data for at least 10 years of daily weather records, digital maps of current rice production area and associated rice yields, and the dominant rice cropping systems

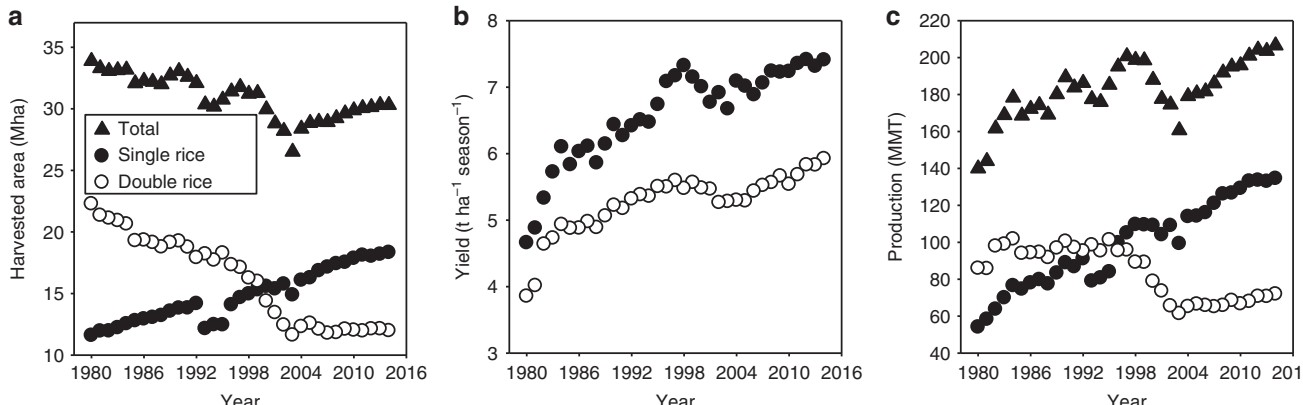

**Fig. 1** Rice production trends in China. Trends in harvested rice area (**a**), rice yield (**b**), and total production (**c**) for single- and double-rice cropping systems during the past 35 years (1980–2014) in China. Note that the yield for double-rice is on a per-harvested area basis so that total annual yield is twice the values shown. Data were obtained from ref. [1]. Mha million hectares, MMT million metric tons. Source data are provided as a Source Data file

and management practices[9,10]. Fifty locations representing 16 climate zones were selected for the reference weather stations (RWS). Because both single- and double-rice systems are prevalent at some of these locations, a total of 847 simulations of rice potential yield were required using a well-validated ORYZA rice simulation model[12] (details on calibration and validation provided in Supplementary Information). These location-specific estimates were then scaled up, based on the proportion of national rice area represented by each location and rice cropping system, to give national estimates of potential yield and current farm yield of 9.8 and 6.8 metric tons per hectare (t ha$^{-1}$) per crop in single- and double- rice systems, respectively, and a national average yield gap of 3.0 t ha$^{-1}$. Hence, current national average rice yield represents 69% of potential yield, which is approaching the 75–80% of the potential yield threshold at which farm yields typically stagnate at regional to national scales due to diminishing returns from further investment in yield-enhancing technologies and inputs[13].

Evaluating yields by climate zone identifies large differences in potential yield among regions and rice cropping systems. For example, estimated potential yields ranged from 8.6 to 10.8 t ha$^{-1}$ across climate zones and rice systems, while current farm yields varied from 5.2 to 8.8 t ha$^{-1}$ (Fig. 2). Farm yields were highest in central regions for both single- and double-rice systems. Year-to-year variability in potential yield was small for both systems as indicated by a temporal coefficient of variation (CV) of 8%, which is typical of the high yield stability found in grain production systems with a reliable supply of irrigation water[14]. In double-rice systems, potential yields of early- and late-season rice crops were similar, with a national average of 9.0 t ha$^{-1}$ for both crops. Although national potential yield of single-rice was 14% greater than that of double-rice per season (10.3 versus 9.0 t ha$^{-1}$), total annual potential yield from double-rice was 18 t ha$^{-1}$ because two crops are produced each year from the same field.

Larger spatial variation in average current farm yields (CV = 14%) than in potential yields (CV = 8%) resulted in a wide range of yield gaps, from 18% to 41% of potential yield across the 16 climate zones evaluated. Current national farm yields for single-rice (7.4 t ha$^{-1}$) and double-rice per season (5.9 t ha$^{-1}$) were 72% and 66% of the potential yields estimated for each system, respectively. Hence, yield gaps of single-rice systems are very close to the 75–80% potential yield threshold at which farm yields tend to stagnate, whereas yield gaps in double-rice systems are considerably below this threshold.

**Potential to increase rice production and be sufficient.** Assuming the exploitable yield gap is estimated by the difference between current farm yield and 80% of potential yield (hereafter called the exploitable yield ceiling), exploitable yield gaps for each of the two crops in double-rice systems are 44% greater than for single-rice (1.3 versus 0.9 t ha$^{-1}$ per season, Table 1). Annual per hectare increase in rice production from closing exploitable yield gaps would be three-fold greater for double-rice than for single-rice (2.6 versus 0.9 t ha$^{-1}$). Correcting for the larger current production area of single-rice gives a total potential increase in rice production of about 16 MMT from double-rice and 15 MMT from single-rice with the closure of exploitable yield gaps on all current rice area (Fig. 3a). Taken together, a scenario of closing exploitable yield gaps on existing rice area would increase national rice production by 15% (+31 MMT, Table 1) compared to current rice production of 206 MMT (average of 2013–2015)[1].

But achieving yields that are 80% of potential yield requires large inputs of fertilizer nutrients and aggressive use of pest control measures to minimize yield losses caused by diseases, insects, and weeds in these continuous rice systems. Precise timing of these inputs with regard to stage of crop development is also required, which means greater investment in labor and expertise to monitor crop status and make tactical modifications to field management during the growing season in response to weather and expected yield levels. Such intensive management may not be economically justifiable if marginal costs of the additional inputs, labor, equipment, and decision-support tools do not cover expected returns, which appears to be the case in California where irrigated rice yields have stagnated at 76% of potential yield[15]. Hence, if the exploitable yield ceiling for profitable rice production is only 75% of potential yield, increased production capacity would be 8% of current production (+16 MMT, Fig. 3a).

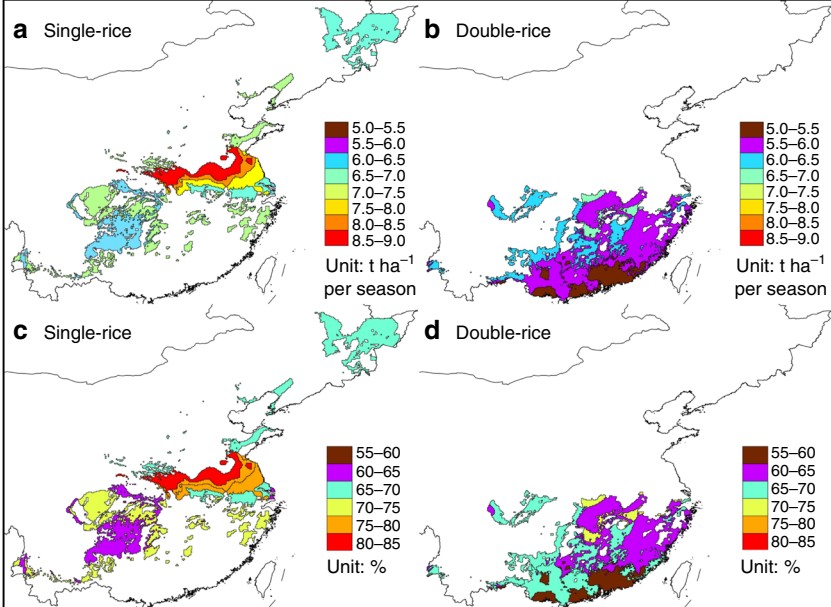

**Fig. 2** Current farm yields and yield gap in China. Current farm yields as absolute values (**a**, **b**) or as a percentage of potential yields (**c**, **d**). All values are reported on a per-harvest basis and are mapped at the climate zone spatial scale. Note that there may be several climate zones within areas showing the similar current farm yield, and thus having the same color in these figures. Source data are provided as a Source Data file

**Table 1 Yields and production of single- and double-rice systems in China and total national production under four scenarios projected to 2030**

| Scenarios | Rice system | Yield (t ha$^{-1}$ season$^{-1}$) | Production (MMT) | Total production (MMT) | Total production compared with demand of 217 MMT in 2030 (MMT) |
|---|---|---|---|---|---|
| S1 | Single-rice | 7.4 | 135.4 | 206 | −11 |
| | Double-rice | 5.9 | 71.0 | | |
| S2, 80% $Y_p$ ceiling | Single-rice | 8.3 | 150.5 | 224 | 7 |
| | Double-rice | 6.1 | 73.6 | | |
| S3, 75% $Y_p$ ceiling | Single-rice | 7.8 | 141.2 | 215 | −2 |
| | Double-rice | 6.1 | 73.6 | | |
| S4, 75% $Y_p$ ceiling | Single-rice | 7.8 | 141.2 | 219 | 2 |
| | Double-rice | 6.5 | 77.9 | | |

S1: Farm yields stagnate at current levels to 2030. S2 and S3: Rates of yield gain follow current trajectories based on regression of national rice yields versus year since 1985 to present for single- and double-rice to 2030 (Supplementary Fig. 6) and an exploitable yield ceiling that is 80% (S2) or 75% (S3) of potential yield ($Y_p$). S4: Rates of yield gain in double-rice increase to the current yield growth rate of single-rice (an increase from 0.03 to 0.05 t ha$^{-1}$ per year per season) and an exploitable yield ceiling that is 75% of $Y_p$. In all four scenarios, there is no change in rice production area for each rice system, which is consistent with recent land use trends as explained in the text. Source data are provided as a Source Data file

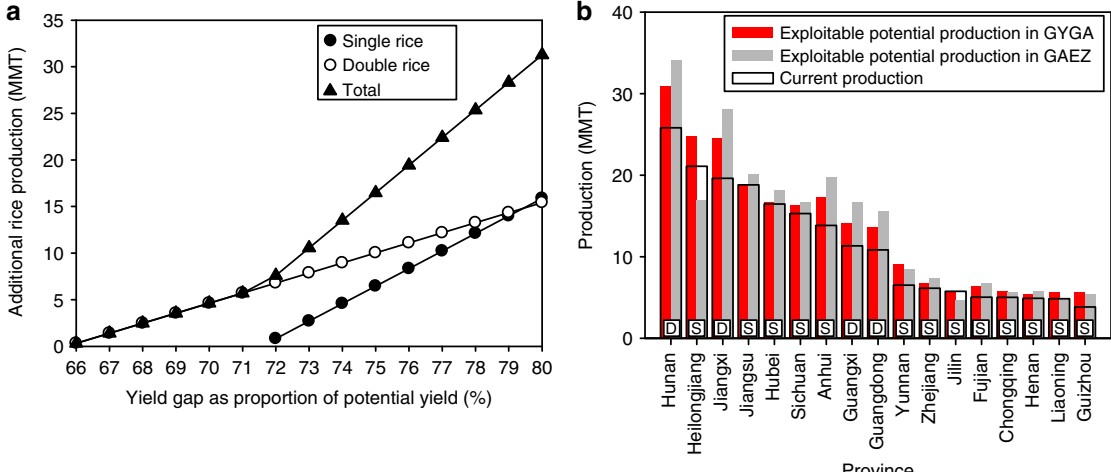

**Fig. 3** Rice production in different cropping systems and provinces. **a** Additional rice production resulting from yield gap closure (expressed as % of potential yield) in single- and double-rice systems assuming no change in harvested area for each system and a maximum exploitable yield ceiling that is 80% of potential yield. **b** Current annual rice production and exploitable potential production for each major rice-growing province as estimated using protocols developed by the Global Yield Gap Atlas (GYGA) or by the Global Agro-Ecological Zones Model version 3.0 (GAEZ v3.0). Capital letters at bottom of each province production bar designate provinces in which double- (D) or single-rice (S) dominate. Exploitable potential production for each province is calculated as the product of provincial average rice planting area of 2013–2015 and the exploitable yield whereby all rice farmers achieve yields that are 80% of potential yield. The current production for each province is based on 2013–2015 average. MMT million metric tons. Source data are provided as a Source Data file

To evaluate plausible future scenarios of rice self-sufficiency, we use a projected total rice demand of 217 MMT in 2030 based on the average estimates of three studies[16–18]. In the four scenarios evaluated, we assume that total rice production area remains constant to 2030, which is consistent with recent trends for double-rice systems (Fig. 1), and also for central and northern provinces where single-rice systems prevail (justification for this assumption in Supplementary Materials and associated Supplementary Figs. 2, 7). Scenario 1 assumes farm yields remain at current levels to 2030. Scenarios 2 and 3 assume that current modest growth rates in rice yields (0.03 and 0.05 t ha$^{-1}$ per year in double- and single-rice, respectively, Supplementary Fig. 6) continue to 2030 and impose an exploitable yield ceiling of either 80% (scenario 2) or 75% (scenario 3) of potential yield (Table 1). By assuming the continuation of current rates of yield gain, we are assuming that improvements in rice cultivars, production practices, and changes in climate and atmospheric $CO_2$ trajectories of the past 30 years will persist. Scenario 4 is the same as scenario 3 but with higher growth rate in yield of double-rice to be equivalent to the yield growth of single-rice, which is modest compared to much faster growth rates experienced during the 1970s and 1980s. Although the higher rate of gain in yield of double-rice in scenario 4 represents a 60% increase compared to the rate of gain in recent decades, we believe such an increase is possible given the fact that double-rice systems have a larger exploitable yield gap than for single-rice systems and would benefit from greater research investment focused on closing this larger yield gap.

Import of 11 MMT would be required to meet projected 2030 rice demand under scenario 1, whereas scenario 2 with an exploitable yield ceiling 80% of potential yield results in a 7 MMT surplus (Table 1), which is equivalent to 11% of current global rice trade. But given concerns about the substantial environmental pollution associated with current intensive crop production practices[19], a scenario with a 75% exploitable yield ceiling is perhaps more appropriate because it would require less fertilizer

input and less aggressive pest control measures. Under this less intensive production scenario, China would be nearly self-sufficient with a modest annual import requirement of 2 MMT. Strategic investment in provinces and rice systems with greatest yield gaps, however, could produce a small surplus even with the more conservative exploitable yield ceiling. For example, the greatest yield gaps are found in double-rice systems in general (Fig. 2), and in some provinces where single-rice dominates. Indeed, seven of the 17 provinces in which rice is a major crop account for 78% of the potential increase in rice supply if all rice farmers achieved yields that reach the exploitable yield ceiling (Fig. 3b). Disaggregation of exploitable production per region shows that the prioritization of crop management improvement interventions and sustainability measures in double-rice systems in general and single-rice systems in three provinces (Heilong-jiang, Anhui, and Yunnan provinces) where yield gaps are relatively large, would provide greatest potential to ensure rice self-sufficiency in China.

Taken together, the larger yield gap and slower yield growth rate suggests greater potential to accelerate yield growth of double-rice, assuming targeted investment in that goal. Raising the rate of yield growth of double-rice to that of single-rice would lead to a rice production surplus of 2 MMT by 2030 with a conservative exploitable yield ceiling of 75% of potential yield. Reducing labor requirements and associated costs in double-rice also would be an important objective of the research and development portfolio focused on these systems.

**Bottom-up versus top-down approaches to estimate yield gaps**. Previous studies have used a top-down spatial framework based on meso-scale grids (roughly 100 km²) into which weather and current crop yield data, obtained from databases at much coarser spatial scales, are interpolated to fit the smaller grid size[20–23]. Likewise, these previous studies do not provide estimates of potential yield as distinguished by different rice cropping systems. In these top-down studies, potential yield is estimated by either the 90th percentile of current farm yields[20], or by a generic crop model not specific for rice[20–23] and therefore not validated for the ability to estimate rice potential yield across the wide variation of rice-growing environments in China.

Comparison of estimates using our bottom-up approach with those using the top-down framework of Global Agro-Ecological Zone (GAEZ) protocols[23] shows a modest difference in national potential production of 17 MMT by 2030, which represents 8% of current national production, and a relatively large fraction (27%) of current global rice trade (documentation of comparison provided in Supplemental Materials and Supplementary Table 8). And while estimates of national potential yield by the two approaches are also in relatively close agreement (10.5 t ha⁻¹ for GAEZ versus 9.8 t ha⁻¹ as reported here), potential production estimates at the provincial level (Fig. 3b) and the estimates of potential yield and yield gap at climate zone scale (Supplementary Fig. 8) differ markedly. For example, in two of the provinces GAEZ potential production estimates are well below current rice production (2013–2015 average). Moreover, in five climate zones in central and southern China, representing 43% of total Chinese rice production, GAEZ estimates a potential yield 18% greater than reported here, and in the two northernmost rice-growing climate zones the GAEZ estimate is 22% less than our estimate (based on data shown in Supplementary Fig. 8). Hence, the relatively close agreement in potential yield estimates at national scale occurs by chance and masks large differences in estimates at finer spatial scales. Such large differences at provincial and climate zone scales between the two approaches would result in very different research and development priorities that seek to focus investments on regions and cropping systems with greatest opportunities for increasing rice production.

## Discussion

That small changes in projected Chinese rice production would have a large impact on global rice trade highlights the need for accurate estimates of potential production on existing rice area. Hence, the spatial resolution of our yield-gap analysis by climate zone, province, and cropping system is essential to adequately inform policies and strategic plans for food security and land use. For this reason our analysis (i) distinguishes between the two major rice cropping systems, (ii) utilizes a rice simulation model that has been rigorously validated for the ability to estimate potential yield across the major rice-growing regions in China (Supplementary Materials, Supplementary Figs. 3–5), (iii) relies on at least 10 years of measured weather data (Supplementary Materials, Supplementary Fig. 1), and (iv) employs a bottom-up scaling protocol validated for capacity to reproduce crop perfor-mance across large variation in climate[9,10].

Our scenario analysis about rice self-sufficiency is based on the assumption that rice yield growth rate was the same across the whole country for each single- and double-rice systems. While our scenario assessment does not account for expected climate change, a number of studies have shown that the magnitude of climate change and associated impact on rice yields is highly uncertain[24–28]. Given the relatively small change in climate for the first half of the century[27,28], together with the short timeframe of our assessment (15 years, from 2016 to 2030), as well as the associated uncertainty about the magnitude of change in climate factors influencing potential yield[24–27], we believe the assessments of rice potential yield provided in our study are both robust and policy-relevant. Likewise, our assessment of future rice produc-tion is optimistic as it assumes that future conversion of prime rice land for residential, industrial, and recreational purposes will be compensated by the addition of new rice land area with a similar level of productivity[5]. If our assumptions on climate change and rice land fall short of reality, this will put additional pressure to accelerate the rate of rice yield gain on existing rice land.

Crop price supports and other types of subsidies to promote self-sufficiency in food production are not considered sound agricultural policies due to high costs, market distortions, and reduced incentives for innovation and efficiencies[29]. In contrast, Clapp[30] argues that populous countries like China may benefit from maintaining self-sufficiency or near self-sufficiency, in production of their primary staple food crops. Hence, for a country like China, reliance on imports for even a small portion of total rice demand represents such a large fraction of global rice trade that pressure from Chinese rice purchases can influence global rice prices in ways that might lead to higher food prices and reduced access, not only for China but also for other coun-tries that import rice. Such pressures are of particular concern in years when global rice supply falls short due to major drought or flooding in rice exporting countries. The good news is that the scenarios evaluated in this study indicate that China has sub-stantial flexibility to maintain self-sufficiency in rice without increasing cropland area devoted to rice production. Indeed, by strategic targeting of investments in research and development, it may be possible to maintain self-sufficiency with a net reduction in rice cropland area, especially if the rate of yield gain can be accelerated in double-rice systems.

## Methods
**Protocols for yield-gap assessment and spatial upscaling**. For irrigated crops such as rice in China, the yield gap represents the difference between current farm yields and the potential yield when the crop is grown without limitations from

nutrient deficiencies, insect pests, or diseases[7]. To estimate yield gaps we followed protocols developed by the GYGA, which utilize primary, location-specific data to the extent possible[9] and a robust upscaling framework to estimate yield gap at larger levels of spatial aggregation such as climate zones (CZ), regions, and national scales[10]. These protocols have been rigorously evaluated for their ability to estimate yield gaps using Australian rainfed wheat as a test case[31]. All underpinning data and protocols are available on the GYGA website (http://www.yieldgap.org/). The GYGA protocols are based on the climate zonation scheme developed by Van Wart et al.[32] that is delineated by three variables: (i) growing degree days, which determine the potential length of the crop growing season, (ii) annual aridity index, which provides an estimate of water supply as the ratio between rainfall and potential evapotranspiration, and (iii) temperature seasonality, which distinguishes between temperate and tropical climates. One hundred and thirty-nine CZ are delineated in China using this climate zonation scheme.

Briefly, a digital map of rice crop area distribution (SPAM map, with $10 \times 10$ km grid-cell resolution)[33] was superimposed on the CZ map to identify weather stations located in areas with greatest density of rice production area (Supplementary Fig. 1). Buffer zones of 100-km radius surrounding each weather station, were clipped following van Bussel et al.[10], so that their borders fall within the same CZ. Weather stations were selected in sequence starting with the station and associated buffer zone with largest rice area and continuing until ca. 50% of national rice area was covered by the selected weather station buffer zones. Earlier work has shown that the inclusion of additional weather stations to achieve greater coverage of rice area does not improve national estimates of yield gap[34]. Selected weather stations are hereafter referred to as RWS. Following this approach, 50 RWS were selected containing 48% of national harvested rice area within their associated buffer zones. The 50 RWS are located in 16 CZ, which together contain 85% of national rice area (Supplementary Fig. 1). Details about selected RWS locations and the dominant rice cropping system within each RWS buffer zone are provided in Supplementary Table 1.

**Simulating rice potential yield.** The ORYZA rice model has been used to simulate rice growth and development of different rice cultivars across a wide range of climatic conditions in Asia, Africa, and USA[15,34–36]. Briefly, ORYZA rice model is a daily-step model that simulates rice phenology, canopy leaf area growth, dry matter accumulation and partitioning, and grain yield formation. Simulation of crop phenology requires calibration of four parameters: development rates for juvenile (DVRJ), photoperiod-sensitive (DVRI), panicle development (DVRP), and reproductive phases (DVRR). Photoperiod sensitivity is assumed negligible, which is consistent with evidence for modern high-yield rice cultivars used in China. Daily rate of canopy $CO_2$ assimilation is calculated from daily incoming radiation, temperature, and leaf area index. Dry matter production is simulated based on the balance between daily $CO_2$ assimilation and respiration. Total biomass is partitioned into various organs depending on allocation rates calibrated for different developmental stages. Potential yield in this model is performed by setting the subroutine of PRODENV = "POTENTIAL" and NITROENV = "POTENTIAL", which basically assumes that the crop grows without water and nutrient limitations and without incidence of pathogens, insect pests, and weeds. In this study, we used the most recent version of this model (ORYZA v3[12]). Although the model still has limitations in simulating rice growth under stress environment, e.g., low temperature, drought, and nitrogen-deficient, etc., it performs well in simulating potential yield under non-stress environment[12,15]. A detailed description of model structure and parameterization can be found in Bouman et al.[36]. Because potential yield is location-specific, the model requires specification of long-term daily weather records and crop management practices for each RWS.

Simulation of annual potential yield and temporal variability in potential yield due to year-to-year variation in weather using the ORYZA rice model requires daily weather records, including maximum and minimum temperature, wind speed, relative humidity, precipitation, and solar radiation. For irrigated cropping systems, 10 years of weather data are sufficient for robust estimation of weather-related temporal variation in potential yield[34]. We obtained daily weather data for 11 years (2004–2014) for all weather variables (except solar radiation) from the National Meteorological Information Center of the China Meteorological Administration. We obtained solar radiation data from the NASA-POWER database (https://power.larc.nasa.gov/). Previous studies have shown that crop yield simulations based on NASA-POWER solar radiation are in close agreement with simulations based on ground-measured radiation across a wide range of environments and regions[37]. Erroneous and/or missing information in weather data were screened following the quality control methods described in van Wart et al.[34] and Grassini et al.[9].

Crop management data required for simulation of potential yield include sowing date, plant density (i.e., number of plants per hill and number of hills per m$^2$), and phenological duration of dominant cultivars. For each RWS and associated buffer zone, we obtained information about the predominant rice cultivars and current yield from local experimental data, and publications reporting agronomic field research conducted at those sites. The selected RWS were grouped into six regions according to CZ and dominant cultivar characteristics following Duan et al.[38,39], and management practices within the same region were assumed to be similar (Supplementary Fig. 2). For example, single-rice systems dominate in the northeast, north, central and southwest regions, while double-rice systems

dominate in the central and south regions. In each region, one widely planted rice cultivar was used for single-crop systems, while two cultivars were used for double-rice systems, one for the early and the other for the late cropping season. A large number of new rice cultivars are released every year and immediately planted by farmers; parameterizing and simulating all of them would have been unfeasible. In this study, we selected the dominant high-yield modern cultivars with broad adaptation and good tolerance to pathogens and insect pests within each region. These cultivars were selected based on the rice variety database (http://www.ricedata.cn/variety/), published research papers, and recommendations from regional rice experts in China.

Since ORYZA rice model was developed for tropical areas, the model was calibrated in the present study to account for the climate conditions and dominant cultivars grown in China. The crop model parameters for an elite rice cultivar in China named Huanghuazhan was carefully calibrated as a standard crop file using experimental data from a 2-year (2012–2013) experiment performed in Huaqiao township (30°06′ N, 115°45′ E), Wuxue county, Hubei province, China. Huanghuazhan is the most common inbred rice cultivar planted in central and south China and is been widely grown across 7 major rice-producing provinces (http://www.ricedata.cn/variety/) with high and stable yield, good quality, and wide adaptability. Although Huanghuazhan is an inbred cultivar, the yield is comparable with hybrid cultivars[40]. Sowing date in the experiment was May 10th, which is typical in central China, and the seedlings were transplanted 26–31 days after sowing. Plant density was 28 hills per m$^2$ with 2–3 plants per hill. Fertilizers were applied at rates of 195 kg N ha$^{-1}$, 33 kg P ha$^{-1}$, and 125 kg K ha$^{-1}$. Fertilizer-N was split into several applications: 36% as basal, 23% at tillering, 23% at panicle initiation, and 18% 1 week after panicle initiation. Fertilizer-P was applied in one single basal application while fertilizer-K was applied 30% before sowing, 40% at tillering, and 40% 1 week after panicle initiation. Insects, diseases, and weeds were periodically controlled using pesticides to avoid biomass and grain yield losses. Leaf area index, and dry matter of leaves, stems, and panicles were measured at different phenological stages.

For the other cultivars, model coefficients were calibrated using phenology, yield and biomass data from studies published in recent years (after 2005) in which crops received near-optimal management, that is, with non-limiting water and nutrient supplied and crops kept free of pathogens, insect pests, and weeds. In those studies involving multiple treatments, we used the highest yield measured across treatments or average yield among treatments if the treatment effect was not significant. Despite our efforts to derive the same set of genetic coefficients for the same rice cultivar irrespective of site or season, it was not possible to portray differences in yield and phenology between late and early season for one of them in one region (Teyou 582 in south region in double-rice systems). Hence, separate coefficients were derived for the early and late season for this cultivar. A similar approach has been followed in calibrating ORYZA in previous studies to portray differences between contrasting environments[41].

Cultivar-specific parameters were derived following standard procedures for the ORYZA rice model, and related information is provided in Supplementary Table 2. Parameterization of crop characteristics was obtained by two utility programs in ORYZA rice model called *drate(v2).exe* and *param(v2).exe*. The *drate(v2).exe* is a program to determine the phenology develop rate of a given variety, and the *param (v2).exe* is a program to estimate crop parameters such as assimilate partitioning, specific leaf area, and non-structure C&N translocation, etc.[12,36]. In our study, program *drate(v2).exe* was used to determine the phenology development rate by using the phenological stages and growth duration of local dominant cultivars. Program *param(v2).exe* was used to estimate the fraction of dry matter partitioned to shoot, the shoot dry matter partitioned to leaves, stems, and panicles at different phenological stages, and the fraction of carbohydrates allocated to stems that is stored as reserves by using the measured dry matter of organs at different phenological stages of local dominant cultivars. See details of calibrated parameters in Supplementary Tables 3–4.

We calibrated and validated the model using independent datasets. We used experimental data from one site-year in each experiment for calibration, and used the other site-year(s) for validation (hereafter referred to as validation 1). We performed an additional validation (hereafter referred to as validation 2) using published yield, biomass and growth duration data for the cultivars included in the calibration to supplement validation 1. These data were not used for model calibration because of the lack of detailed information required for model calibration or because there was not enough information to ensure that the crops were managed to reach potential yield; hence, it was expected that a number of the observed yields reported in these studies would fall below the simulated potential yield. Details about experimental data for model calibration and validation can be found in Supplementary Tables 2–4 and Supplementary Fig. 3.

Degree of association and agreement between simulated and observed variables was assessed by the coefficient of determination ($r^2$), root mean square error (RMSE), and RMSE expressed as percentage of the observed mean (RMSE$_n$), which were calculated as follows:

$$r^2 = \left( \frac{n(\sum xy) - (\sum x)(\sum y)}{\left[ n\sum x^2 - (\sum x)^2 \right]\left[ n\sum y^2 - (\sum y)^2 \right]} \right)^2 \quad (1)$$

$$RMSE = \left[ \left( \sum (x - y)^2 / n \right) \right]^{0.5} \qquad (2)$$

$$RMSE_n = \left[ \left( \sum (x - y)^2 / n \right) \right]^{0.5} / M_{mean} \times 100\% \qquad (3)$$

where $x$ and $y$ represent the simulated and observed values, and $n$ represents the number of paired values. $r^2$ close to 1 and RMSE and $RMSE_n$ close to 0 indicate a good agreement between simulated and observed values.

Observed grain yields across experiments ranged from 8.4 to 14.4 t ha$^{-1}$ and were in close agreement with simulated values after model calibration as indicated by relatively low RMSE 0.89 t ha$^{-1}$ (validation 1), which represented 9% of the mean observed yield (Supplementary Fig. 5). Likewise, validation 1 indicated close agreement between simulated and observed aboveground dry matter and growth duration, with low RMSE for shoot biomass (0.80 t ha$^{-1}$) and growth duration (5 days). In the validation 2 database, yield, total biomass, and growth duration ranged from 6.5–11.6 t ha$^{-1}$, 10.8–20.7 t ha$^{-1}$, and 106–185 days, respectively. Results in validation 2 also showed reasonable agreement between simulated yield and observed yield, with RMSE representing 15% and 16% of the mean observed yield and biomass, respectively. As expected, a number of observed yields were above the 1 to 1 line (Supplementary Fig. 5c). Growth duration was in close agreement as indicated by the low RMSE of 5%. To summarize, reasonable agreement between simulated and observed yields using two different datasets gives confidence that the calibrated ORYZA rice model is robust at reproducing potential yield across the wide range of climates and rice cropping systems in China.

We performed a review of previous efforts in calibrating and/or validating rice simulation models in studies aiming to estimate rice potential yield at regional and national levels (Supplementary Table 5). In most of those studies, model coefficients were neither calibrated nor validated using local experimental data; instead, potential yield was simulated using default coefficients derived somewhere else or published in the literature. In some studies, the model was calibrated and/or evaluated only for one or few varieties and/or using data from few site-years. An exception was two studies using the database from the Agrometeorological Experimental Stations of the Chinese Meteorological Administration (CMA)[26,42]. The CMA database has advantages given its large number of sites and information on rice crop phenology and yield. However, the data are collected from farmer fields that do not receive management conducive to expression of potential yield. Likewise, the database only has data on rice phenology and yield at harvest time, which are insufficient for a robust model calibration and/or evaluation. Hence, the detailed model calibration and evaluation performed for a large number of cultivars across a wide range of environments in China is a clear strength of our study compared with previous efforts aiming to estimate potential yield at regional and national levels.

All data sources and uncertainties, and associated quality control measures to mitigate possible biases, are listed in Supplementary Table 6. In all cases, data sources fall in the so-called tier 1 of data availability/quality for crop yield gap analysis described in Grassini et al.[9].

**Current farm yield and yield gaps**. Available yield data from the 5 most recent years (2010–2014) were retrieved for each of the counties that overlap with the selected RWS from national and provincial statistical bureaus. However, county-level farm yield data are not accurate for a number of reasons, and the magnitude of inaccuracy varies across counties and is difficult to predict[43,44]. In contrast, provincial level farm yield data are more reliable and accurate because a combination of different methods are used, including remote sensing and ground truthing[45]. Hence, using county-level data without adjustment to be consistent with the provincial-level yield data gives inaccurate yield gap estimates when results are aggregated to larger spatial scales. To adjust county-level yield data in a province, we increased or decreased farm yields by an equivalent percentage so that the weighted average farm yields across all RWS within that province equaled the provincial official average yield of 2010–2014. However, we did not adjust farm yield of counties in provinces for which there was little difference (within ±5%) between the official provincial farm yield and the upscaled provincial farm yields following GYGA protocols. Average current yield was calculated as the average year over the 5-year (2010–2014) time period to represent yield in the current season.

The dominant cropping system (single- or double-rice) was identified for each RWS buffer zone and used as the basis for simulating potential yields and for estimating yield gaps. In some parts of central China and in northern provinces, only a single-rice crop is grown each year because the growing season is too short for double-rice. Hence, the yield gap for each RWS where a single-rice crop is grown was calculated as the difference between the single-rice potential yield and the current farm yield. In south and some parts of central China, farmers practice a double-rice cropping system. However, county-level data provide only the average yield for the two crops. But the area of early- and late-season rice crops is almost identical at 19% and 20% of total rice area, respectively, based on the estimates from ref. [1]. We therefore simulated yields of early- and late-rice crops separately, and used the average potential yield of the two crops to calculate yield gaps for each RWS where double-rice systems dominate.

To upscale potential yield, current farm yield, and yield gap estimates from RWS to larger spatial scales, weighted averages for each variable were calculated by the proportional contribution of rice area within each spatial unit contributing to the spatially aggregated value at the CZ or national scale[10].

**Estimating rice production potential**. Current farm yields tend to stagnate when they reach 75–80% of potential yield (called the exploitable yield ceiling) due to diminishing returns from investment in additional production inputs and effort as yields approach the potential yield ceiling[13,15,34]. Hence, prospects for increasing rice production at any spatial scale are best indicated by the exploitable portion of the yield gap, which is the difference between current farm yields and 75% or 80% of potential yields estimated at the CZ and national scales. All else equal, achieving yields that are 80% of potential yield requires greater input of nutrients and more aggressive pest control measures than production at 75% of potential yield. Given concerns about the substantial environmental pollution associated with current intensive crop production practices[19], scenarios with either an 80% or 75% exploitable yield ceiling were evaluated. Additional exploitable production potential of single- versus double-rice systems was estimated by the difference between 75% or 80% of potential yield and current farm yields for each rice cropping system over the total national production area.

**Future scenarios**. Rice production scenarios to 2030 were evaluated based on the following assumptions:

(i)　Yield would stagnate at current levels to 2030 or follow current trajectories based on regression of national rice yields versus year for single- and double-rice (Supplementary Fig. 6).

(ii)　Amount of cropland devoted to rice production remains unchanged at the level of 2011–2014 average, which is consistent with current trajectories based on the fitted trends of observed harvested area (Supplementary Fig. 6). We plotted harvested area and yield against year (1985–2014) for different rice cropping systems by linear function or by a linear-segment piecewise function using SigmaPlot 10.0 (Systat. Software, Inc., San Jose, CA, USA), as shown in Supplementary Fig. 6.

Linear function:

$$y = at + b \qquad (4)$$

where $t$ is year, $y$ is single-rice harvested area or single-, double-rice yield.

Two-linear-segments piecewise function:

$$t_1 = \min(t), \text{ which is the year 1985} \qquad (5)$$

$$t_2 = \max(t), \text{ which is the year 2014} \qquad (6)$$

$$y = \begin{cases} \frac{a(T_1 - t) + b(t - t_1)}{T_1 - t_1}, & t_1 \leq t \leq T_1 \\ \frac{b(t_2 - t) + c(t - T_1)}{t_2 - T_1}, & T_1 \leq t \leq t_2 \end{cases} \qquad (7)$$

where $t$ is year, $y$ is total- or double-rice harvested area, and $T_1$ is the breakpoint year. The $r^2$ for the linear regressions of total-, single-, and double-rice harvested area are 0.73, 0.88, and 0.93, and the $r^2$ for the linear regressions of single- and double-rice yield are 0.78 and 0.71. All estimated parameters were statistically significant (Student's $t$-test, $P < 0.0001$).

The linear regression for single-rice yield:

$$y = 0.0488\,t - 90.788 \qquad (8)$$

The linear regression for double-rice yield:

$$y = 0.0263\,t - 47.255 \qquad (9)$$

(iii)　While a similar regression for single-rice area shows a continued linear increase through 2015, disaggregation of these data by province shows that most of the recent increase occurred in Heilongjiang province, and to a lesser extent in Jilin province while rice production area in remaining single-rice provinces has remained constant or even decreased in recent years (Supplementary Fig. 7). Moreover, recent government policies have advocated for reduced rice area in Heilongjiang due to limited water resources.

(iv)　Total rice demand in 2030 is 217 MMT from the average of three recent studies[16–18], which is 5.3% higher compared with current rice production of 206 MMT (average of 2013–2015 from NBSC[1] to represent current rice production).

(v)　An exploitable yield ceiling of either 80% or 75% of potential rice yield.

**Comparison of studies assessing rice production scenarios**. We cross-validated our scenario assessment results by comparing them with results from recent studies assessing future rice for China at country level (Supplementary Table 7). The goal of some of the studies was not to estimate rice self-sufficiency per se but they reported data that allowed us to assess it. These studies followed diverse approaches as basis for their assessments. We found that the results from our scenario assessment focusing on rice systems and regions fall within the range of projected self-sufficiency ratios with scenarios considering other factors e.g., socio economic,

technology development, and climate change etc., which adds confidence on the results reported here.

**Comparison of yield differences by GAEZ and GYGA methods.** Global Agro-Ecological Zones Model version 3.0 (GAEZ v3.0)[23] was used for comparison of yield-gap analyses. Compared to the GYGA up-scaling method, a down-scaling method is used in GAEZ v3.0 such that both weather and agricultural crop production data (e.g., current yields and crop production area) from much coarser spatial scales are interpolated into 5 arc-min grid cells (roughly 100 km$^2$). National estimates are then estimated by aggregation of data from all grids in which there is rice production area. In contrast to our GYGA approach, which utilized provincial-level data on proportion of rice area under either single- or double-rice systems to estimate area of each system within RWS station buffer zones, GAEZ assigned one rice cropping system to each 5 arc-min grid cell by matching growth cycle and temperature requirements of rice with length of time available for crop growth[23].

In GAEZ the "Agro-climatic exploitable yields with high input level" data layer is defined as potential climatic yield with optimal management practices ($Y_{P\_GAEZ}$), which is a proxy for potential yield as estimated using GYGA methodology ($Y_{P\_GYGA}$). The map of Agro-climatic exploitable yields with high input level for irrigated rice (map) was downloaded on May 26, 2018 from GAEZ website: http://gaez.fao.org/.

We calculated $Y_{P\_GAEZ}$ at CZ, province and country levels and compared them with $Y_{P\_GYGA}$ at those same levels of spatial upscaling using the following approach. First, each GAEZ grid cell (5 arc-min) in $Y_{P\_GAEZ}$ map was superimposed with the irrigated rice harvested area from the SPAM map[33] to assign a harvested rice area to each grid cell. Second, each grid cell was assigned to a GYGA CZ based on the CZ map. The two processes were performed in ArcGIS 10.2. Weighted $Y_p$ values based on harvested area were then estimated at the CZ, province and country level. Potential yield in GAEZ was calculated:

$$Y_p = \frac{\sum_{i=1}^{n} Y_{pi} \times A_i}{\sum_{i=1}^{n} A_i} \qquad (10)$$

where $Y_p$ is potential yield, and $A$ is harvested area. For $Y_p$ at CZ level, $i$ is a grid cell and $n$ is the number of grid cells within a CZ. For $Y_p$ at province level, $i$ is a grid cell and $n$ is the number of grid cells within a province. For single-rice $Y_p$, $i$ is a CZ and $n$ is the number of CZ within single-rice system. For double-rice $Y_p$, $i$ is a CZ and $n$ is the number of CZ within double-rice system. For $Y_p$ at country level, $i$ is a cropping system and $n$ is the number of cropping systems within a country. $Y_g$ in GAEZ ($Y_a$ as a percent of $Y_p$) was calculated:

$$Y_{g\_GAEZ}(\%) = \frac{Y_{a\_GYGA}}{Y_{p\_GAEZ}} \times 100\% \qquad (11)$$

where $Y_a$ in GYGA was used to calculate $Y_g$ for GAEZ because the $Y_a$ in GAEZ is outdated (from the year 2000).

## Data availability

All data generated or analyzed during this study are available within this paper and its Supplementary Information files. The source data underlying Table 1, Figs. 1–3, Supplementary Figs. 1–8, and Supplementary Table 8 are provided as a Source Data file. A reporting summary for this article is available as a Supplementary Information file.

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

## Acknowledgements

The authors would like to thank Dr. David Lobell for commenting on an earlier version of the paper and suggesting the comparison of bottom-up versus top-down yield-gap analysis approaches. This research was supported by the earmarked fund for the National Key Research and Development Program of China (No. 2016YFD0300210), China Agriculture Research System (CARS-01-20), the Program of Introducing Talents of Discipline to Universities in China (the 111 Project no. B14032), the Program for Changjiang Scholars and Innovative Research Team in University (IRT1247), and the Daugherty Water for Food Global Institute at University of Nebraska-Lincoln.

## Author contributions

All authors conceived the study and wrote the paper. N.D., H.Y. and P.G. performed the statistical analysis and simulations. All authors contributed to editing the paper.

## Additional information

**Competing interests:** The authors declare no competing interests.

