## [Peer Review File · Nature Communications]

Reviewers' comments:

Reviewer #1 (Remarks to the Author):

The paper "'To be or not to be: Prospects for rice self-sufficiency in China" is well written with high significance to current debates about food security. It contains some notable features:

- combines large scale assessment with a relative good coverage of diverse local management conditions (double versus single cropping for instance), which is often lacking in large scale assessments
- Thoughtful development of scenarios with respect to the socio-economic conditions
- Comparisons of two approaches to assess yield gaps on larger scale: GYGA (Global Yield Gap Atlas, bottom-up) and GAEZ (Global Agro-Ecological Zone, a more top-down approach)

However, I have two major concerns:

1) A key problem is the treatment of climate change. The authors state:

"Although climate change will have an impact on rice yield potential and regional production potential as evaluated in these scenarios, the magnitude of climate change by 2030 is projected to be relatively small compared to the impact during the second half of this century." As basis they give Rosenzweig (2014) Proc. Natl. Acad. Sci. USA 111, 3268-3273.

Rosenzweig et al. presented a coarse assessment without testing of the models or including realistically local management conditions. However, the authors of this submitted paper stressed the importance of bottom-up studies taking local management into account. Hence, using this reference for their statement in this paper is a bit contradicting. Furthermore, at least and that is also what is found in Rosenzweig the CO₂ effect will play a role for a C3 plant such as rice. So climate change receives by far too little attention by this paper. There are some discussion how this should be taken into account taking the uncertainty of climate model simulations into account (see Corbeels et al. 2018) <https://doi.org/10.1016/j.agrformet.2018.02.026>) and the uncertainty associated with the CO₂ effect. In addition, we face regional climate changes due to economic activities, i.e. solar dimming (see for instance Wang and Wild (2016) <https://doi.org/10.1002/2016GL071009>). So, this aspect of a changing climate is missing, and if a paper claims to provide an outlook for 2030 this needs to be addressed, especially for a journal with a wide audience including journalists and policy-makers. Hence, possible limitations/shortcomings need to become very clear

2) Model testing

In L385 the authors discuss the results of the model validation exercise, and conclude finally in Line 395 that the "model is robust at reproducing potential yield across the wide range of climates and rice cropping systems in China." I challenge this statement when looking at Figure 3. There is too little data available to say 'across China'. I count 15 yield data points, for shoot biomass and growth duration there is even less. Especially, for the growth duration the results don't look convincing, for instance Yongyou12. Also the data was obviously not detailed enough to provide a more accurate, respectively detailed cultivar description. In the main text the authors underline the importance of canopy development (so leaves) for potential yield. Parameters associated with this property were not modified according to Table 2. But it would be interesting to see especially how this would affect potential yield (that could be linked to scenarios where actually also breeding plays a role, which is not taken into account in this paper). Furthermore, there is a need to add a section (just supplement) with a short description of the ORYZA v3 model (including for instance limitations in regard to for instance lodging) and a table that at least a well-trained person in the model can understand how the model was set-up (is there for instance a planting density factor in the model?).

Acknowledging the efforts of the authors to develop a bottom-up assessment and the significance of the topic, I would recommend reject with the option of re-submission. However, a re-submitted version requires a wider testing of the model (i.e. more data from more sites; including a clear description of field trials in the appendix) and secondly climate change needs to be addressed.

Minor comments:

L39-46: Well written section on the significance of this study: Self-sufficiency versus importing rice

L60: How is potential yield depending on 'annual crop rotation sequence'?

L61-64: what is missing is the radiation use efficiency or in other terms the efficiency of the crop to transfer the intercepted light into biomass

L61: When we talk about the year 2030, we need to consider as well an increase of CO₂, so a likely increase in efficiency of the rice to produce biomass?

L99 (and also later): yield potential or potential yield? it is just semantics, but please stay consistent with the terms.

L124: now attainable yield. Why not just continuing to use potential yield? We are talking about irrigated systems. So it should not make a difference...

L664 Figure 6: please explain abbreviations in the captions

Figure 1: explain MMT and Mha in the caption.

L229: A good example actually exists for maize and wheat in China: Zhang, W.; Cao, G.; Li, X.; Zhang, H.; Wang, C.; Liu, Q.; Chen, X.; Cui, Z.; Shen, J.; Jiang, R.; Mi, G.; Miao, Y.; Zhang, F.; Dou, Z. Closing yield gaps in China by empowering smallholder farmers. *Nature* 2016, 537, 1–16, doi:10.1038/nature19368. In my opinion, this could be an important reference to discuss in this context.

Reviewer #2 (Remarks to the Author):

Review:

Title: To be or not to be: Prospects for rice self-sufficiency in China

What are the major claims of the paper?

This paper reports on the analysis of the trends of China rice production and on foresight analysis on its directions to 2030 with the constraints of the resources availability. The concern of rice self-sufficiency in China is of global food security concern and the authors were highlighting clearly this to introduce the importance of their work. The work gave as the authors claim an explicit spatial distribution of rice potential of production in the major rice growing areas of the country that allows identification of areas with high potential of yield improvement with larger yield gaps and areas reaching a plateau of productivity with yield approaching the exploitable yield.

Are they novel and will they are of interest to others in the community and the wider field?

The yield gaps analysis done by the authors followed the existing framework of Global Yield Gap Atlas with the novelty to include temporal evaluation and future climate scenarios. Their results are of interest of any perspective in rice production analysis and trends. Their methodology in using statistical data and standard farmers practices to estimate actual yields are of interest as existing methodology using farmers survey to define farmers practices are very tedious and time consuming in addition of the large variability of reported farmers yield with minimum association with the variability of their practices.

The conclusions are giving perspectives on how China can maintain self-sufficiency without increasing cultivated area by focusing on research and development in reducing yield gap. These are not novel finding but confirming the need of more initiative in yield gap reduction as reported for instance by Stuart et al., 2017.

However the quantification in the potential importation needed and surplus are certainly new considering the authors only take into account biophysical factors of weather and land availability. The authors should add then some cross check with foresight analysis done by other authors if

their findings are consistent with valid scenarios considering socio economic factors.

Is the work convincing, and if not, what further evidence would be required to strengthen the conclusions?

The work is well written and clear with a straight forward conclusion supported by methodology that innovates in the use of data from different sources. It is an approach that clearly abstract many uncertainties in the evaluation of rice cropping systems at scale which It will be great if the authors mentioned some of these uncertainties and as well the limitation of the validity of their approach.

Among them the authors specifically ignore the contribution of the variety in the estimation of potential yield, thus by assuming the 6 six regions for parameterization of the models, they assumed that the six regions have no variability in varieties used. If that the case please provide references justifying this assumption.

It was not clear as well for which year the current season was set, as yield gap has temporal variability please state the considered season for the analysis. Please refer as well then the baseline of comparison to the 2030 projection and how the baseline was computed.

I have enjoyed reading the paper and I have appreciated the ability of the authors to infuse modeling work into simple message but I am afraid without clear statement of the limitation and the factors that the study did not account for in the interpretation of the results and the conclusion they may create misleading information for general use and perspectives.

I congratulate the authors for their work in trying to simplify as possible the systematic data consuming methodology in model use and they have as well create here a valuable data set in weather data and reported yields data for Chinese growing rice areas that can be now referenced for further studies.

I have few concerns that I would like the authors to consider:

1. Please use yield per ha per season in the labelling of the graph.
 2. Please present the map of 127 AEZ covering china and the 16 CZ you have identified covering the 85% of rice growing areas with the delimitation of the 6 regions for farmers practices. These steps are the main foundation of your spatial analysis and it misses in the report.
 3. The references for the data collected for model calibration is less from international references for clarity please provide in the table 2 the date of the experiments source of data for calibration reported in these references.
 4. I understand you did not consider the variety in the calibration but please state in the material and methods your assumption on the variety for the yield potential simulation (what parameters were calibrated from which data and from which crop file (variety) is your standard reference values).
 5. Please state as well in the scenarios your consideration of variety change as the continuous linear yield gain may come from new variety adoption and improvement in addition of any favorable condition of rice production with climate change to 2030.
- Again the authors did not mention any uncertainties in their source of data and results, except the correction that they made in the county yield level to match the $\pm 5\%$ of the provincial yield level to permit downscaling analysis for higher resolution. Please state these sources and any quantification if possible in your results and your conclusion.

Reviewer #3 (Remarks to the Author):

This study provides a modeling study to assess the rice production and yield gaps in China at present and in 2030. The study claims it to be the first “high-resolution spatial analysis of rice production potential in China”. The study concluded that by reducing the yield gaps China’s rice production may be self-sufficient for its demand in 2030.

The study has an overall reasonable logic, and the storyline is clear. So if the evidence provided (i.e. the modeling results) can pass the scrutiny, I would think that this work would be a significant and interesting contribution. This being said, I think the current results need to provide much more technical details on how their modeling results were generated with more validations. In the current form, the calibration and validations of the model are very thin. For a crop modeler like me, the technical details of the modeling part here are hand-waving at most. Considering the whole study almost all relies on the modeling results, I don’t think the manuscript has reached the rigorousness and standard for Nature Communications.

Specifically, the authors claimed that they did the model calibration, but only very brief information was provided in the supplementary materials. Specifically, in line 360 they said “The param(v2).exe was used to estimate crop parameters such as assimilate partitioning among organs, leaf area index, and specific leaf area at different phenological stages of local dominant cultivars.” First, the authors should list all the calibrated parameters and the variables that they used to calibrate, instead of using “such as”. Second, my read here is that the authors calibrated (1) biomass allocation to different crop component, (2) LAI, and (3) specific leaf area (SLA), for different pheno stages and for different rice cultivars. The first two, i.e. biomass and LAI, should be the model state variables rather than model parameters, and the SLA can be a model parameter. Thus, the above statement sounds to me as from someone who does not even understand model outputs and model parameters. In fact, at least a couple of parameters in ORYZA needs to be calibrated, usually related to the rate of pheno stage development, biomass allocation, stress. Please be very specific to list all these parameters and what model output variables (also the correspondent observations) they use in the calibration process, and provide details on how the calibration is done.

The authors only chose 9 sites to calibrate and validate their modeling results. First, for each site, I ask the authors to provide the information of all the calibrated parameters’ value in a table to ensure the reproducibility of your modeling results by others. Second, the authors should provide a comparison of some critical model outputs and how they compared with observations, such as LAI, biomass, and pheno-stage simulation (see the next paragraph for more details). Considering the study is so dependent on scaling these 9 site results to the whole national rice growing region, such a justification is required.

Furthermore, Extended Figure 3 is the only model validation result here. However, in each panel the authors only provided 9 points, each corresponding to one site. So the questions are: whether these results are your in-sample validated results (i.e. the simulated results after you calibrated your models using the observed data), or out-of-sample validation for different years? Does each site only have one year of data? The correct way is to use some data to calibrate your model, and then apply your calibrated model to different year’s weather data to do the out-of-sample test. You should report both in-sample and out-of-sample comparison results. The current Ext Fig 3 seems to me are way less sufficient and could not convince me your modeling results. I also question whether the authors have the sufficient data for the current modeling study/calibration.

By doing a quick search of rice crop modeling of China in google scholar, I can find many literatures. Thus I am not sure how valid for the claim that this is the first study of “high-resolution spatial analysis of rice production potential in China”. Furthermore, I am not sure what the “high-resolution” really means here – I assume it means that spatially explicit and fine resolution. However, in reality, the model only conducted the simulation at the site level, and then scale up to the national level. If “high-resolution’ wants to be justified, the authors should at least provide

some "high-resolution" yield maps instead of the current Figure 2.

I also suggest the authors avoid any exaggerated use of languages. The title is a typical example. Please keep it plain, specific, and use supported substance to make your claim.

Responses to reviewers' comments on "To be or not to be: Prospects for rice self-sufficiency in China" by Deng *et al.*

We appreciate the reviewers' comments and their overall positive view about our study. We have considered all their comments in revising the MS and provided detailed responses to them (see below in red font). Many parts of the MS have been revised in response to this feedback. We also provided more information about the model evaluation section and explained the relevance of our scenario assessments despite expected climate change and continued genetic improvement of rice varieties. We believe that the revised MS has been improved a lot as a consequence of these revisions, and we thank the three reviewers for their constructive input.

Reviewer #1 (Remarks to the Author):

The paper "To be or not to be: Prospects for rice self-sufficiency in China" is well written with high significance to current debates about food security. It contains some notable features:

- combines large scale assessment with a relative good coverage of diverse local management conditions (double versus single cropping for instance), which is often lacking in large scale assessments
- Thoughtful development of scenarios with respect to the socio-economic conditions
- Comparisons of two approaches to assess yield gaps on larger scale: GYGA (Global Yield Gap Atlas, bottom-up) and GAEZ (Global Agro-Ecological Zone, a more top-down approach)

ANSWER (1-1; reviewer #1 - answer 1): We appreciate the comments from reviewer #1 about our paper, and his/her support for publication.

However, I have two major concerns:

1) A key problem is the treatment of climate change. The authors state:

"Although climate change will have an impact on rice yield potential and regional production potential as evaluated in these scenarios, the magnitude of climate change by 2030 is projected to be relatively small compared to the impact during the second half of this century." As basis they give Rosenzweig (2014) Proc. Natl. Acad. Sci. USA 111, 3268-3273. Rosenzweig *et al.* presented a coarse assessment without testing of the models or including realistically local management conditions. However, the authors of this submitted paper stressed the importance of bottom-up studies taking local management into account. Hence, using this reference for their statement in this paper is a bit contradicting. Furthermore, at least and that is also what is found in Rosenzweig the CO₂ effect will play a role for a C₃ plant such as rice. So climate change receives by far too little attention by this paper. There are some discussion how this should be taken into account taking the uncertainty of climate model simulations into account (see Corbeels *et al.* 2018) <https://doi.org/10.1016/j.agrformet.2018.02.026>) and the uncertainty associated with the CO₂ effect. In addition, we face regional climate changes due to economic activities, i.e. solar dimming (see for instance Wang and Wild (2016) <https://doi.org/10.1002/2016GL071009>). So, this aspect of a changing climate is missing, and if a paper claims to provide an outlook for 2030 this needs to be addressed, especially for a journal with a wide audience including journalists and policy-makers. Hence, possible

limitations/shortcomings need to become very clear

ANSWER (1-2): we agree with Reviewer #1 that Rosenzweig (2014), as well as other studies about the impact of climate change on crop yields, followed a very coarse methodology, with little attention paid to the cropping system context in which crops are grown, or to the straightforward agronomic adaptation measures that farmer can take to deal with climate change (such as modification of planting dates and crop maturities). However, our intention when citing this paper was NOT to compare our simulated results against those reported by this (or other) previous studies on climate change impact on yield. Instead, the point we make in citing Rosenzweig (2014) is that the effect of climate change on crop yields is expected to be relatively small during the first half of the 21st century (2000-2050) compared with the second half (2050-2100). Following reviewer's comment, we added other references to strengthen this point in the revised MS (see L. 153 of revised MS). Given the relatively small change in climate for the first half of the century, together with the relatively short timeframe of our assessment (15 years, from 2016 to 2030), as well as the associated uncertainty about the magnitude of change in climate factors influencing yield potential, we believe the assessments of rice yield potential provided in our study are both robust and policy-relevant. For example, the overall change of 0.3-0.7 °C in mean annual temperature for the period 2016-2035 (IPCC, 2014; reference 21) would be equivalent to an annual change of 0.02-0.05 °C, which can likely be mitigated through tactical changes in varietal maturity and sowing dates (reference 22-24). Likewise, as an irrigated crop grown in standing water, the actual temperature in the rice canopy is somewhat attenuated from the highs and lows of air temperature per se. Based on Reviewer #1 comments, however, we provide more information about the impact of climate change on our assessment (see L. 151-157 of revised MS).

2) Model testing

In L385 the authors discuss the results of the model validation exercise, and conclude finally in Line 395 that the “model is robust at reproducing potential yield across the wide range of climates and rice cropping systems in China.” I challenge this statement when looking at Figure 3. There is too little data available to say ‘across China’. I count 15 yield data points, for shoot biomass and growth duration there is even less. Especially, for the growth duration the results don't look convincing, for instance Yongyou12. Also the data was obviously not detailed enough to provide a more accurate, respectively detailed cultivar description. In the main text the authors underline the importance of canopy development (so leaves) for potential yield. Parameters associated with this property were not modified according to Table 2. But it would be interesting to see especially how this would affect potential yield (that could be linked to scenarios where actually also breeding plays a role, which is not taken into account in this paper). Furthermore, there is a need to add a section (just supplement) with a short description of the ORYZA v3 model (including for instance limitations in regard to for instance lodging) and a table that at least a well-trained person in the model can understand how the model was set-up (is there for instance a planting density factor in the model?).

ANSWER (1-3): Thanks for these comments. In response, we added more detail and data points for model validation (See new Extended Data Figs. 3-5 and Extended Data Table 2-4 of revised MS). Basically, we first used experimental data of one year/site for calibration, and the other

year/site for validation in the same paper (referred to as 'validation 1'). Second, we expanded our model evaluation to test model ability to reproduce potential yield by performing an additional model validation ('validation 2') with actual yield data from field experiments representing additional years and locations using same cultivars used in our study. Hence validation 2 supplements validation 1, but data source was different from calibration and validation 1. The details about experimental data for model calibration and validation can be found in the new Extended Data Tables 2-4, Extended Data Fig. 3, and L. 457-479, 494-504 of revised MS). Results on model calibration and validation are shown in Extended Data Figs. 4-5, and L. 514-529 of revised MS.

The reviewer stressed the relatively small number of observations used for model evaluation. We believe that more important than the number of observations is the range of environments where these data were collected and the quality of the experiments. Because our goal was to simulate yield potential for major rice producing areas in China, we selected experiments managed to achieve yields without limitations from water, nutrients or pests, and which covered the range of rice cultivars and growing environments in China for model calibration, with yields ranging from 8.4 to 14.4 t/ha, and crop cycle from 120 to 177 days. Moreover, we added a map showing the experimental sites in the Supplementary Section of revised MS (see new Extended Data Fig.3). Unfortunately, published studies in which experiments were managed to achieve yields near the yield potential ceiling are relatively few, so the quantity of data of relevance to our validations is limited. Still, we believe our model calibration and evaluation are considerably more thorough than previous studies performing regional crop production assessments. Indeed, we find very little effort devoted to calibrating and/or evaluating rice models in previous published papers assessing national rice potential production, with the calibration typically limited to a few regions and/or varieties. We have added text and a new table to highlight the strength of our model calibration and evaluations compared with previous studies (see Extended Data Table 5 and L. 530-546 of revised MS).

This reviewer is correct in his/her observation about rice cultivar Yongyou12. Because we did not use Yongyou12 as one of the cultivars for simulating yield potential for any region in China, we decided to remove it from the analysis. In response to this reviewer's comment about yield gains derived from plant breeding, we have added text to note that by extending the current rate of gain in rice yields to 2030 in our scenario analyses, we are assuming continued improvements in rice varieties as has occurred in past decades. Hence, the impact of genetic crop improvement is included in our scenario analyses (see L. 166-168 of revised MS).

We appreciate reviewer's comment about being transparent in relation with model structure and parameterization. We have added a section with a short description and limitations of the ORYZA v3 model (see L. 395-417 of revised MS), a number of tables listing all calibrated model parameters (see new Extended Data Table 3-4 of revised MS) and extending the associated section describing model calibration to make sure readers and other modelers understand how the calibration was performed (see L. 494-504 of revised MS).

Acknowledging the efforts of the authors to develop a bottom-up assessment and the

significance of the topic, I would recommend reject with the option of re-submission. However, a re-submitted version requires a wider testing of the model (i.e. more data from more sites; including a clear description of field trials in the appendix) and secondly climate change needs to be addressed.

ANSWER (1-4): We appreciate reviewer #1 positive comments about our efforts and the significance of this study. We have added more details and data points for model validation, elaborated on the technical details of the modeling work and field trials, and discussed the climate change issue in greater detail. We believe that the paper is now more rigorous at describing the model evaluation and addressing climate change so that reviewers and readers feel more confident about the results.

Minor comments:

L39-46: Well written section on the significance of this study: Self-sufficiency versus importing rice

Thanks

L60: How is potential yield depending on 'annual crop rotation sequence'?

ANSWER (1-5): Growth duration of each crop cycle tends to be shorter in systems producing two crops per year (i.e. double-rice), which means that yield potential of each crop cycle is less than in single-rice systems which typically use rice cultivars of longer growth duration. However, total annual rice production per hectare is greater in double-rice despite the lower yield per crop cycle than in single-rice. We changed "annual crop rotation sequence" into "crop growth duration" (see L. 71 of revised MS). We apologize for the confusion with terminology.

L61-64: what is missing is the radiation use efficiency or in other terms the efficiency of the crop to transfer the intercepted light into biomass

ANSWER (1-6): We believe that whereas temperature regime and light interception vary greatly site to site and year to year, the conversion efficiency is nearly constant for a given crop species. So no change was made in revised MS.

L61: When we talk about the year 2030, we need to consider as well an increase of CO₂, so a likely increase in efficiency of the rice to produce biomass?

ANSWER (1-7): Following reviewer's suggestion, we added text elaborating on this of revised MS (see L. 166-168 of revised MS).

L99 (and also later): yield potential or potential yield? it is just semantics, but please stay consistent with the terms.

ANSWER (1-8): Following reviewer's comments, we chose one term (potential yield) and stick to it in the entire MS.

L124: now attainable yield. Why not just continuing to use potential yield? We are talking about irrigated systems. So it should not make a difference...

ANSWER (1-9): We referred to the yield exploitable by farmers using best available, cost-effective technology which usually leads to ca. 80% of the yield potential. This was explained in the original MS (see L. 131-132, 585-591 of revised MS). We changed the term “attainable” into “exploitable” in revised MS. The key point here is that farmers strive to maximize profit and not yield per se, which is why yields at a regional or national level tend to stagnate at about 80% of potential yields as found in several studies.

L664 Figure 6: please explain abbreviations in the captions

ANSWER (1-10): Done (see Extended Data Fig. 8 of revised MS).

Figure 1: explain MMT and Mha in the caption.

ANSWER (1-11): Done (see Figs. 1, 3, and Extended Data Figs. 6, 7 of the revised MS).

L229: A good example actually exists for maize and wheat in China: Zhang, W.; Cao, G.; Li, X.; Zhang, H.; Wang, C.; Liu, Q.; Chen, X.; Cui, Z.; Shen, J.; Jiang, R.; Mi, G.; Miao, Y.; Zhang, F.; Dou, Z. Closing yield gaps in China by empowering smallholder farmers. Nature 2016, 537, 1–16, doi:10.1038/nature19368. In my opinion, this could be an important reference to discuss in this context.

ANSWER (1-12): Thanks for the suggestion. However, we feel that the approach of empowering smallholder farmers for closing yield gaps is not very relevant to the main conclusion of our manuscript. Therefore, we had a difficulty to include this reference during the revision.

Reviewer #2 (Remarks to the Author):

Review:

Title: To be or not to be: Prospects for rice self-sufficiency in China

What are the major claims of the paper?

This paper reports on the analysis of the trends of China rice production and on foresight analysis on its directions to 2030 with the constraints of the resources availability. The concern of rice self-sufficiency in China is of global food security concern and the authors were highlighting clearly this to introduce the importance of their work. The work gave as the authors claim an explicit spatial distribution of rice potential of production in the major rice growing areas of the country that allows identification of areas with high potential of yield improvement with larger yield gaps and areas reaching a plateau of productivity with yield approaching the exploitable yield.

ANSWER (2-1): We appreciate reviewer #2 positive comments about our study.

Are they novel and will they be of interest to others in the community and the wider field?

The yield gaps analysis done by the authors followed the existing framework of Global Yield Gap Atlas with the novelty to include temporal evaluation and future climate scenarios. Their results are of interest from any perspective in rice production analysis and trends. Their methodology in using statistical data and standard farmers practices to estimate actual yields are of interest as existing methodology using farmers survey to define farmers practices are very tedious and time consuming in addition of the large variability of reported farmers yield with minimum association with the variability of their practices.

The conclusions are giving perspectives on how China can maintain self-sufficiency without increasing cultivated area by focusing on research and development in reducing yield gap. These are not novel findings but confirming the need of more initiative in yield gap reduction as reported for instance by Stuart et al., 2017.

However the quantification in the potential importation needed and surplus are certainly new considering the authors only take into account biophysical factors of weather and land availability. The authors should add then some cross check with foresight analysis done by other authors if their findings are consistent with valid scenarios considering socio economic factors. Is the work convincing, and if not, what further evidence would be required to strengthen the conclusions?

ANSWER (2-2): We have found a number of studies that attempt to assess rice production scenarios for China based on climate change or economic considerations and we have listed them in a new table (see Extended Data Table 7 and L. 636-645 of revised MS). We found that the results from our scenario assessment focusing on rice systems and regions fall within the range of projected self-sufficiency ratios with scenarios considering other factors like socio economic, technology development, and climate change etc. This 'cross-validation' adds confidence to the findings of our study and we thank reviewer #2 for this great idea.

The work is well written and clear with a straight forward conclusion supported by methodology that innovates in the use of data from different sources. It is an approach that clearly abstracts many uncertainties in the evaluation of rice cropping systems at scale which it will be great if the authors mentioned some of these uncertainties and as well the limitation of the validity of their approach.

Among them the authors specifically ignore the contribution of the variety in the estimation of potential yield, thus by assuming the 6 six regions for parameterization of the models, they assumed that the six regions have no variability in varieties used. If that the case please provide references justifying this assumption.

ANSWER (2-3): Thanks for the suggestions. We also acknowledge that there are lots of cultivars used within a region. However, we couldn't include all the cultivars used by farmers due to lack of data and, perhaps more importantly, the fast varietal turnover would make our results obsolete very quickly. Instead, our approach was to select, for each region, the most common

high-yield modern cultivar, with broad adaptability and good pest resistance. We elaborated on this point in the revised MS (see L. 446-452, 457-479). We also added a table listing all data sources and associated uncertainties and limitations and quality control measures implemented to reduce biases (see Extended Data Table 6 and L. 547-550 of revised MS). We note that all our data sources fall into the so-called tier 1 of data availability/quality for crop modeling described by Grassini et al (2015)⁹.

It was not clear as well for which year the current season was set, as yield gap has temporal variability please state the considered season for the analysis. Please refer as well then the baseline of comparison to the 2030 projection and how the baseline was computed.

ANSWER (2-4): *Current season in our study was considered as the average year over the 2010-2014 time period. We rephrased the text to make this point clear (see L. 567-568 of revised MS). For the baseline of comparison to the 2030 projection, we used total rice production of 206 MMT (an average data of 2013-2015 from NBSC¹). We rephrased the text following reviewer's suggestion (see L. 632-634 of revised MS).*

I have enjoyed reading the paper and I have appreciated the ability of the authors to infuse modeling work into simple message but I am afraid without clear statement of the limitation and the factors that the study did not account for in the interpretation of the results and the conclusion they may create misleading information for general use and perspectives.

We believe we have addressed all comments made by Reviewer #2 of revised MS (see previous responses). We are thankful to Reviewer #2 because his/her comments helped us prepare a much better version of the MS, in which assumptions and datasets are more transparent.

I congratulate the authors for their work in trying to simplify as possible the systematic data consuming methodology in model use and they have as well create here a valuable data set in weather data and reported yields data for Chinese growing rice areas that can be now referenced for further studies.

Again, we appreciate reviewer #2 positive comments about our study and support for publication.

I have few concerns that I would like the authors to consider:

1. Please use yield per ha per season in the labelling of the graph.

ANSWER (2-5): *Done (see revised Figs. 1-2 and Extended Data Fig. 6 and related text in the revised MS).*

2. Please present the map of 127 AEZ covering china and the 16 CZ you have identified covering the 85% of rice growing areas with the delimitation of the 6 regions for farmers practices. These steps are the main foundation of your spatial analysis and it misses in the report.

ANSWER (2-6): Done (see revised Extended Data Fig. 2 of the revised MS).

3. The references for the data collected for model calibration is less from international references for clarity please provide in the table 2 the date of the experiments source of data for calibration reported in these references.

ANSWER (2-7): Done (see revised Extended Data Table 2 of the revised MS).

4. I understand you did not consider the variety in the calibration but please state in the material and methods your assumption on the variety for the yield potential simulation (what parameters were calibrated from which data and from which crop file (variety) is your standard reference values).

ANSWER (2-8): See our previous response on this in ANSWER (2-3). Our standard reference values are from Huanghuazhan crop file. We added the detailed information about it in Extended Data Table 2-4 and L. 457-479 of revised MS.

5. Please state as well in the scenarios your consideration of variety change as the continuous linear yield gain may come from new variety adoption and improvement in addition of any favorable condition of rice production with climate change to 2030.

ANSWER (2-9): Thanks for the comments, we added it in L. 166-168 of revised MS.

Again, the authors did not mention any uncertainties in their source of data and results, except the correction that they made in the county yield level to match the +5% of the provincial yield level to permit downscaling analysis for higher resolution. Please state these sources and any quantification if possible in your results and your conclusion.

ANSWER (2-10): See our previous response on this in ANSWER (2-3).

Reviewer #3 (Remarks to the Author):

This study provides a modeling study to assess the rice production and yield gaps in China at present and in 2030. The study claims it to be the first “high-resolution spatial analysis of rice production potential in China”. The study concluded that by reducing the yield gaps China’s rice production may be self-sufficient for its demand in 2030.

The study has an overall reasonable logic, and the storyline is clear. So if the evidence provided (i.e. the modeling results) can pass the scrutiny, I would think that this work would be a significant and interesting contribution. This being said, I think the current results need to provide much more technical details on how their modeling results were generated with more validations. In the current form, the calibration and validations of the model are very thin. For a crop modeler like me, the technical details of the modeling part here are hand-waving at most.

Considering the whole study almost all relies on the modeling results, I don't think the manuscript has reached the rigorousness and standard for Nature Communications.

ANSWER (3-1): we appreciate reviewer #3 positive comments about the significance and relevance of our study. We have elaborated on the technical details of the modeling work and we believe that the paper is now more rigorous at describing the model evaluation so that reviewers and readers feel more confident about the results. See our detailed responses below.

Specifically, the authors claimed that they did the model calibration, but only very brief information was provided in the supplementary materials. Specifically, in line 360 they said "The param(v2).exe was used to estimate crop parameters such as assimilate partitioning among organs, leaf area index, and specific leaf area at different phenological stages of local dominant cultivars." First, the authors should list all the calibrated parameters and the variables that they used to calibrate, instead of using "such as". Second, my read here is that the authors calibrated (1) biomass allocation to different crop component, (2) LAI, and (3) specific leaf area (SLA), for different pheno stages and for different rice cultivars. The first two, i.e. biomass and LAI, should be the model state variables rather than model parameters, and the SLA can be a model parameter. Thus, the above statement sounds to me as from someone who does not even understand model outputs and model

parameters. In fact, at least a couple of parameters in ORYZA needs to be calibrated, usually related to the rate of pheno stage development, biomass allocation, stress. Please be very specific to list all these parameters and what model output variables (also the correspondent observations) they use in the calibration process, and provide details on how the calibration is done.

ANSWER (3-2): In the ORYZA v3 model, the program param(v2).exe is a program to estimate crop parameters such as assimilate partitioning, specific leaf area, and non-structure C&N translocation, etc³⁹⁻⁴⁰. In our study, this program was used to estimate the fraction of dry matter partitioned to shoot, the shoot dry matter partitioned to leaves, stems and panicles at different phenological stages, and the fraction of carbohydrates allocated to stems that is stored as reserves by using the dry matter of organs at different phenological stages of local dominant cultivars. We apologize for any confusion about this program. We rephrased the section describing model calibration to make sure readers and other modelers understand how the calibration was performed (see L. 483-504 of revised MS) and we added a table listing all calibrated model parameters (see Extended Data Table 3-4 of revised MS).

The authors only chose 9 sites to calibrate and validate their modeling results. First, for each site, I ask the authors to provide the information of all the calibrated parameters' value in a table to ensure the reproducibility of your modeling results by others. Second, the authors should provide a comparison of some critical model outputs and how they compared with observations, such as LAI, biomass, and pheno-stage simulation (see the next paragraph for more details). Considering the study is so dependent on scaling these 9 site results to the whole national rice growing region, such a justification is required.

ANSWER (3-3): Following his/her suggestion on adding an evaluation of other critical model outputs, we added the information of calibrated parameters in Extended Data Table 3-4, and the comparisons of biomass (total and/or per organ) and growth duration for a number of sites in Extended Data Figs. 4-5 of the revised MS. See also our previous response in ANSWER (1-3) about how we performed model evaluation and made justifications.

Furthermore, Extended Figure 3 is the only model validation result here. However, in each panel the authors only provided 9 points, each corresponding to one site. So the questions are: whether these results are your in-sample validated results (i.e. the simulated results after you calibrated your models using the observed data), or out-of-sample validation for different years? Does each site only have one year of data? The correct way is to use some data to calibrate your model, and then apply your calibrated model to different year's weather data to do the out-of-sample test. You should report both in-sample and out-of-sample comparison results. The current Ext Fig 3 seems to me are way less sufficient and could not convince me your modeling results. I also question whether the authors have the sufficient data for the current modeling study/calibration.

ANSWER (3-4): Following reviewer's comments, we added more data points for model validation (See new Extended Data Fig. 3, 5 of the revised MS). We used experimental data of one year/site for calibration, and the other year/site for validation in the same paper. We named the validation as validation 1. Second, we expanded our model evaluation to test model ability to reproduce potential yield by performing an additional model validation with more years and sites for the same cultivars that were calibrated. The data source was different from the database used for calibration and validation 1. The details about experimental data for model calibration and validation can be found in the new Extended Data Tables 2-4 and Extended Data Fig.3-5 of revised MS. Our analysis showed the calibrated parameters were robust for reproducing observed yield in the independent dataset used for model evaluation. Following reviewer's suggestion, we elaborate more on the text to make this point explicit (see L. 494-504 of revised supplementary material).

About quantity of data, see our previous detailed response in Answer (1-3) to Reviewer #1. In summary, we believe our model calibration and evaluation are considerably more thorough than previous studies performing national crop production assessments. Indeed, we find very little effort devoted to calibrating and/or evaluating rice models in previous published papers assessing rice production potential, with or without climate change, with the calibration typically limited to a few regions and/or varieties. We have added text and a new table to highlight the strength of our model calibration and evaluations compared with previous studies (see Extended Data Table 5 and L. 530-546 of revised MS).

By doing a quick search of rice crop modeling of China in google scholar, I can find many literatures. Thus I am not sure how valid for the claim that this is the first study of "high-resolution spatial analysis of rice production potential in China". Furthermore, I am not sure what the "high-resolution" really means here – I assume it means that spatially explicit and fine resolution. However, in reality, the model only conducted the simulation at the site level, and

then scale up to the national level. If “high-resolution’ wants to be justified, the authors should at least provide some “high-resolution” yield maps instead of the current Figure 2.

***ANSWER (3-5):** thanks for the comment. Our intention was to refer to the degree of disaggregation by crop cycle, climate zones, etc. We replaced ‘high-resolution’ by ‘detailed’ (see L. 30 and 78 of the revised MS). We apologize for the confusion with the terminology.*

I also suggest the authors avoid any exaggerated use of languages. The title is a typical example. Please keep it plain, specific, and use supported substance to make your claim.

***ANSWER (3-6):** We improved our text to be plain and more specific. For the title of the paper, we like it but we are open to change if the editor requests us to do so.*

REVIEWERS' COMMENTS:

Reviewer #1 (Remarks to the Author):

Review of the revised paper: "To be or not to be: Prospects for rice self-sufficiency in China"

I had two main concerns with the previous version of this paper:

- i) missing discussion of climate change effects
- ii) not sufficient transparency of modelling calibration/validation, not sufficient testing

About i), the authors have addressed it sufficiently.

About ii), surely, as the authors stated, there is limited data available for doing the necessary calibration/validation. That might be also a key reason why other scholars have not done it yet. However, I acknowledge the effort of the authors to extend this section. It is a huge effort to compile the calibration/validation data from these different sources. Based on the validation results, it appears that the model is indeed able to capture region-specific relative differences in potential yields.

In addition, the calibration/validation process plus the section on model structure has been made much clearer. I believe, based on this readers are able to understand the limitations of the study.

To sum up, in my opinion this study presents necessary novel results for a wide audience, a transparent description of the methodology, and just enough (although on the edge) testing of the model, which all together justifies publications.

Reviewer #2 (Remarks to the Author):

What are the major claims of the paper?

This work is very timely as food security remained a constant challenge globally and country wise. The insight provided by this work is of interest for policy makers for China itself but as well for different countries producers and consumers of rice. This work set an important step in the use of modelling led hypothesis to set priority in research program and in development intervention. It was very important that the authors set clearly the boundaries of these hypothesis and the factors considered as they themselves mentioned that by taking into account the turnover of varieties used the studies may be obsolete. They have followed rigorous scientific approach that can be applied and referenced to as among hypothesis in how country like china an other rice growing countries plan their strategy for self-sufficiency in rice crop in the near future.

The authors concluded that with conservative scenarios in term of land cropped and improvement in crop and in crop management practices, China would need to import about -11 MMT by 2030 to meet the current projected demand of its population. In contrast with significant effort to reduce yield gaps mainly in double rice cropping systems, it may have surplus about 2MMT. These results are based on assumptions that rice is grown in a similar way across the whole country. The alternative of these assumptions were not tested which certainly is off the scope of paper but it will be better for the reader to have a statement of this at the end of the discussion.

Are they novel and will they be of interest to others in the community and the wider field?

The data presented and used by the authors are valuable for modelling community and for foresight analysis at local and global level.

The authors emphasized on the validity of their approach by comparing the results using the GAEZ which may be among the main contribution of this work in innovating in foresight analysis in crop production at local level.

If the conclusions are not original, it would be helpful if you could provide relevant references.

The conclusion is that reducing yield gaps remained a significant option to contribute to food security, which is not original in itself. This work is emphasizing on the effect of reducing yield gaps with the focus and the example of rice crop in China, and they moved further than usual work by quantifying what are the consequences of reducing yield gaps at country level and by presenting the spatial distribution of the variability of these gaps in rice growing areas of the country. The authors have referenced most of the known work on Yield Gaps and they state by themselves in the text that their results are consistent and not different. The originality of this work is the quantification part translates into deficit and surplus of production. And they mentioned they are consistent with other studies using less data.

The authors may need then to highlight their ability to disaggregate this quantification per regions and that they can identify based on the given hypotheses which areas has more potential to prioritize crop management improvement interventions and how much they have potentially of total production gap and in which areas plateau and sustainability measures has to be implemented to maintain areas of production and level of productivity. They mentioned that objective in Line 85 it would be great to have the plausible answer stated in the abstract.

Is the work convincing, and if not, what further evidence would be required to strengthen the conclusions?

By providing details description of their data used and their modelling set up, the authors have provided enough detail to support the robustness of their results. Their assumptions are very strong and abstract but it provides a reference baseline that can be cross checked in time to evaluate what factors drive rice production in China if they are up for self-sufficiency, if not how the factors considered in the scenarios have changed.

On a more subjective note, do you feel that the paper will influence thinking in the field? This work is of interest and I really appreciate that the authors brought in front modelling as a discipline of research that can bring meaningful contribution to push forward research program and as well development planning in the discussion of regional and global food security. It can set then a new reference on how to use crop model. Their results can as well be a reference as I mentioned earlier to evaluate the trends of rice production and the cost of self-sufficiency when it will be associated to econometrics analysis

Please feel free to raise any further questions and concerns about the paper.

The writing of the work has been very improved and their main research objective and methodology is very clear. The title may be off misleading as self-sufficiency is not just about rice but it reflects the dichotomy choice between self-sufficiency and importing rice.

On Line 87 they use the work geography of China rice production, may be landscape will be appropriate.

On line 105 please provide the ref of the rice simulation model, it would be great as the author only use one model to name it at this stage the well validated ORYZA rice model

Line 108 please precise when you refer to seasonal yield or annual yield

Line 122 please add 18t/ha/year

Line 244 I would suggest to have it in the abstract as this is what novel to be communicated by this work in addition of the role of modelling as novel approach

Line 408 please change PRDENV TO PRODENV

Line 415 Bouman et al.

Line 439 and 443 there is Central repeated two times please rename individually the 6 regions as indicated in the fig 2 as Central southwest and central South??? Not sure which central

Line 457 please rephrase that the crop model parameters for the variety Huanghuazhan were calibrated using field experimental data

Extended Table 2 and 3- We have two different parameters for one variety Teyou582- Please check and explain as the two may be grown in two different seasons but the variation in parameters look like they are two different varieties. Please kindly check and update accordingly

Comment on the appropriateness and validity of any statistical analysis, the ability of a researcher to reproduce the work, given the level of detail provided.

I am satisfied with the calibration and validation procedure as the authors provided enough detail that can help to reproduce their work, However I have a little bit of doubt in the computation of the RMSE in the validation set 2 as they have it very low for such large variation within the data points. It was very strange as well that the CV of variation of the actual farms yield was low of 14% Line (124) . I would assume the authors did only keep the maximum yield values. Please restate the sampling procedure of the validation 2 and the source of data for the farm yields.

Reviewer #3 (Remarks to the Author):

I read the authors' responses, and agree that they have made a large effort in addressing my concerns. Thus my prior concerns have been mostly addressed. There is space for further improvement, but I will let the authors pass.

Responses to reviewers' comments on "Prospects for rice self-sufficiency in China" by Deng *et al.*

We thank the reviewers for their positive recommendation on our revised MS for publication in Nature Communications. We also thank them for the additional comments. Our responses are shown below in red. The line numbers indicated in the response letter are the line numbers in the main text in "Simple Markup" or "No Markup" mode.

REVIEWERS' COMMENTS:

Reviewer #1 (Remarks to the Author):

Review of the revised paper: "To be or not to be: Prospects for rice self-sufficiency in China"

I had two main concerns with the previous version of this paper:

- i) missing discussion of climate change effects
- ii) not sufficient transparency of modelling calibration/validation, not sufficient testing

About i), the authors have addressed it sufficiently.

About ii), surely, as the authors stated, there is limited data available for doing the necessary calibration/validation. That might be also a key reason why other scholars have not done it yet. However, I acknowledge the effort of the authors to extend this section. It is a huge effort to compile the calibration/validation data from these different sources. Based on the validation results, it appears that the model is indeed able to capture region-specific relative differences in potential yields.

In addition, the calibration/validation process plus the section on model structure has been made much clearer. I believe, based on this readers are able to understand the limitations of the study.

To sum up, in my opinion this study presents necessary novel results for a wide audience, a transparent description of the methodology, and just enough (although on the edge) testing of the model, which all together justifies publications.

ANSWER: We appreciate the positive comments about the significance of our study and recommendation for publication.

Reviewer #2 (Remarks to the Author):

What are the major claims of the paper?

This work is very timely as food security remained a constant challenge globally and country wise. The insight provided by this work is of interest for policy makers for China itself but as well for different countries producers and consumers of rice. This work set an important step in the use of modelling led hypothesis to set priority in research program and in development intervention. It was very important that the authors set clearly the boundaries of these hypothesis and the

factors considered as they themselves mentioned that by taking into account the turnover of varieties used the studies may be obsolete. They have followed rigorous scientific approach that can be applied and referenced to as among hypothesis in how country like china another rice growing countries plan their strategy for self-sufficiency in rice crop in the near future.

ANSWER: We also thank this reviewer for the positive comments.

The authors concluded that with conservative scenarios in term of land cropped and improvement in crop and in crop management practices, China would need to import about -11 MMT by 2030 to meet the current projected demand of its population. In contrast with significant effort to reduce yield gaps mainly in double rice cropping systems, it may have surplus about 2MMT. These results are based on assumptions that rice is grown in a similar way across the whole country. The alternative of these assumptions were not tested which certainly is off the scope of paper but it will be better for the reader to have a statement of this at the end of the discussion.

ANSWER: Thanks, we have added related text in discussion following reviewer's suggestion (L. 271-273 of revised MS).

Are they novel and will they be of interest to others in the community and the wider field?

The data presented and used by the authors are valuable for modelling community and for foresight analysis at local and global level.

The authors emphasized on the validity of their approach by comparing the results using the GAEZ which may be among the main contribution of this work in innovating in foresight analysis in crop production at local level.

ANSWER: Thanks again for the comments.

If the conclusions are not original, it would be helpful if you could provide relevant references.

The conclusion is that reducing yield gaps remained a significant option to contribute to food security, which is not original in itself. This work is emphasizing on the effect of reducing yield gaps with the focus and the example of rice crop in China, and they moved further than usual work by quantifying what are the consequences of reducing yield gaps at country level and by presenting the spatial distribution of the variability of these gaps in rice growing areas of the country. The authors have referenced most of the known work on Yield Gaps and they state by themselves in the text that their results are consistent and not different. The originality of this work is the quantification part translates into deficit and surplus of production. And they mentioned they are consistent with other studies using less data.

The authors may need then to highlight their ability to disaggregate this quantification per regions and that they can identify based on the given hypotheses which areas has more potential to prioritize crop management improvement interventions and how much they have potentially

of total production gap and in which areas plateau and sustainability measures has to be implemented to maintain areas of production and level of productivity. They mentioned that objective in Line 85 it would be great to have the plausible answer stated in the abstract.

ANSWER: Thanks for the suggestion, we have added text in the end of abstract to highlight the double-rice systems and three provinces in single-rice which have more potential to prioritize crop management improvement interventions in revised MS following reviewer's suggestion (L. 33-39 of revised MS).

Is the work convincing, and if not, what further evidence would be required to strengthen the conclusions?

By providing details description of their data used and their modelling set up, the authors have provided enough detail to support the robustness of their results. Their assumptions are very strong and abstract but it provides a reference baseline that can be cross checked in time to evaluate what factors drive rice production in China if they are up for self-sufficiency, if not how the factors considered in the scenarios have changed.

ANSWER: Thanks.

On a more subjective note, do you feel that the paper will influence thinking in the field? This work is of interest and I really appreciate that the authors brought in front modelling as a discipline of research that can bring meaningful contribution to push forward research program and as well development planning in the discussion of regional and global food security. It can set then a new reference on how to use crop model. Their results can as well be a reference as I mentioned earlier to evaluate the trends of rice production and the cost of self-sufficiency when it will be associated to econometrics analysis

ANSWER: Thanks.

Please feel free to raise any further questions and concerns about the paper. The writing of the work has been very improved and their main research objective and methodology is very clear. The title may be off misleading as self-sufficiency is not just about rice but it reflects the dichotomy choice between self-sufficiency and importing rice.

ANSWER: Thanks. For the title, our primary justification for this paper is to evaluate potential for rice self-sufficiency given the historical national policy in China to maintain self-sufficiency in the past three decades and pressures on land and water resources. Therefore, we would like to keep the title centered on the self-sufficiency issue, and we believe it is appropriate to use "self-sufficiency" in the title.

On Line 87 they use the work geography of China rice production, may be landscape will be appropriate.

ANSWER: We believe that geography is appropriate noun to use in this sentence so no change was made.

On line 105 please provide the ref of the rice simulation model, it would be great as the author only use one model to name it at this stage the well validated ORYZA rice model

ANSWER: We have added the reference for rice simulation model (See L. 124) and kept the term "ORYZA rice model" throughout the main text and supplementary information.

Line 108 please precise when you refer to seasonal yield or annual yield

ANSWER: Done (see L. 127-129 of revised MS).

Line 122 please add 18t/ha/year

ANSWER: "per year" is redundant as we note that total production is "annual". (See L. 144-145 of revised MS)

Line 244 I would suggest to have it in the abstract as this is what novel to be communicated by this work in addition of the role of modelling as novel approach

ANSWER: Thanks for the comments. We have added a sentence in the abstract for the novelty: "A focus on increasing yields of double-rice systems in general, and in three single-rice provinces where yield gaps are relatively large, would provide greatest return on investments in research and development to remain self-sufficient."

Line 408 please change PRDENV TO PRODENV

ANSWER: Done (see L. 349 of revised MS).

Line 415 Bouman et al.

ANSWER: Done (see L. 356 of revised MS).

Line 439 and 443 there is Central repeated two times please rename individually the 6 regions as indicated in the fig 2 as Central southwest and central South??? Not sure which central

ANSWER: Thanks for catching this. Labels were corrected (See new Supplementary Figure 2)

Line 457 please rephrase that the crop model parameters for the variety Huanghuazhan were calibrated using field experimental data

ANSWER: Done (see L. 394-396 of revised MS).

Extended Table 2 and 3- We have two different parameters for one variety Teyou582- Please check and explain as the two may be grown in two different seasons but the variation in parameters look like they are two different varieties. Please kindly check and update accordingly

ANSWER: Despite our efforts to derive the same genetic coefficients for the same rice cultivar irrespective of site or season, it was not possible to portray differences in yield and phenology between late and early season for one of them in one region (Teyou 582 in south region in double-rice cropping system). Hence, separate coefficients were derived for the early and late season for this cultivar. A similar approach has been followed in calibrating ORYZA in previous studies to portray differences between contrast environment⁴¹. We have added some sentences in the revised MS to explain this (L. 417-423 of revised MS).

Comment on the appropriateness and validity of any statistical analysis, the ability of a researcher to reproduce the work, given the level of detail provided.

I am satisfied with the calibration and validation procedure as the authors provided enough detail that can help to reproduce their work, However I have a little bit of doubt in the computation of the RMSE in the validation set 2 as they have it very low for such large variation within the data points. It was very strange as well that the CV of variation of the actual farms yield was low of 14% Line (124). I would assume the authors did only keep the maximum yield values. Please restate the sampling procedure of the validation 2 and the source of data for the farm yields.

ANSWER: We double-checked and the calculations for RMSE and CV are correct (Detailed calculations can be found in the Source Data file). In relation with the small spatial CV: we note that this CV represents variation in average rice yield across climate zones and NOT variation across individual farmer fields. Small spatial CVs are expected for average irrigated rice yield because irrigation buffers against rain-free periods; hence, CVs of irrigated crops are much smaller than for rainfed crops that depend upon water supply from precipitation and stored soil water at planting. Thanks for pointed it out and we added a word "average" before "current yields" to make this sentence clearer (L. 146-148 of revised MS).

Reviewer #3 (Remarks to the Author):

I read the authors' responses, and agree that they have made a large effort in addressing my concerns. Thus my prior concerns have been mostly addressed. There is space for further improvement, but I will let the authors pass.

ANSWER: We thank the reviewer for the positive comments and recommendation for publication.